# A Novel Approach to Deliver Therapeutic Extracellular Vesicles Directly into the Mouse Kidney via Its Arterial Blood Supply

**DOI:** 10.3390/cells9040937

**Published:** 2020-04-10

**Authors:** Mujib Ullah, Daniel D. Liu, Sravanthi Rai, Mehdi Razavi, Jeff Choi, Jing Wang, Waldo Concepcion, Avnesh S. Thakor

**Affiliations:** 1Interventional Regenerative Medicine and Imaging Laboratory, Department of Radiology, Stanford University School of Medicine, Palo Alto, CA 94304, USA; ullah@stanford.edu (M.U.); liudan@stanford.edu (D.D.L.); drrai@stanford.edu (S.R.); merazavi@stanford.edu (M.R.); jc2226@stanford.edu (J.C.); jingw1@stanford.edu (J.W.); 2Biionix (Bionic Materials, Implants & Interfaces) Cluster, Department of Internal Medicine, College of Medicine, University of Central Florida, Orlando, FL 32827, USA; 3Department of Materials Science & Engineering, University of Central Florida, Orlando, FL 32816, USA; 4Department of Surgery, Stanford University, Stanford, CA 94305, USA; wconcepcion@stanfordmed.org

**Keywords:** locoregional delivery, intra-arterial delivery, targeted therapy, microsurgery, acute kidney injury, mesenchymal stromal cells, extracellular vesicles

## Abstract

Diseases of the kidney contribute a significant morbidity and mortality burden on society. Localized delivery of therapeutics directly into the kidney, via its arterial blood supply, has the potential to enhance their therapeutic efficacy while limiting side effects associated with conventional systemic delivery. Targeted delivery in humans is feasible given that we can access the renal arterial blood supply using minimally invasive endovascular techniques and imaging guidance. However, there is currently no described way to reproduce or mimic this approach in a small animal model. Here, we develop in mice a reproducible microsurgical technique for the delivery of therapeutics directly into each kidney, via its arterial blood supply. Using our technique, intra-arterially (IA) injected tattoo dye homogenously stained both kidneys, without staining any other organ. Survival studies showed no resulting mortality or iatrogenic kidney injury. We demonstrate the therapeutic potential of our technique in a mouse model of cisplatin-induced acute kidney injury (AKI). IA injection of mesenchymal stromal cell (MSC)-derived extracellular vesicles (EVs) successfully reversed AKI, with reduced physiological and molecular markers of kidney injury, attenuated inflammation, and restoration of proliferation and regeneration markers. This reproducible delivery technique will allow for further pre-clinical translational studies investigating other therapies for the treatment of renal pathologies.

## 1. Introduction

Kidney diseases contribute a significant morbidity and mortality burden on society: acute kidney injury (AKI) causes 9.5% of in-hospital mortality, and chronic kidney disease (CKD), most commonly due to hypertension and diabetes, has a staggering 14% prevalence in the United States [1]. Progression of both AKI and CKD can ultimately result in end stage renal disease, for which dialysis and kidney transplantation are the only treatment options. These treatment modalities are themselves demanding for patients and fraught with health complications, requiring multiple long visits to the hospital or the need to be on life-long immunosuppression. In addition to AKI and CKD, kidney cancer also contributes a significant burden on society with approximately 60,000 Americans diagnosed with this disease each year [2].

Although there have been significant advances for treating these pathologies, the majority of promising pharmacological and cellular therapies under investigation are administered by conventional intravenous (IV) injection [3,4]. Directly delivering therapeutics to organs, via their arterial blood supply, has many advantages over IV injection, including minimizing systemic side effects, maximizing the therapeutic efficacy at the target site, and avoidance of first pass metabolism in the liver or sequestration by the reticuloendothelial system and lungs. While intra-arterial (IA) delivery of therapeutics is already widely clinically utilized for the treatment of stroke [5], heart attacks [6], and liver cancer [7], there are comparatively few indications for IA delivery of therapeutics for pathologies of the kidney. In part, this can be attributed to limited pre-clinical studies given that performing non-terminal IA injection into the kidneys in small animal models is technically challenging. Indeed, to our knowledge, despite well-documented mice models of AKI, CKD, and orthotropic renal tumors, there is no published methodology to successfully deliver therapeutics into both kidneys in mice, via their arterial supply.

Mesenchymal stromal cells (MSCs) have shown promising results for kidney regeneration in the context of AKI and CKD [4,8]. The therapeutic effect of MSC-based therapies is thought to come from their ability to home to damaged tissue and secrete soluble factors with regenerative properties. Due to the paracrine nature of this mechanism, the proximity of infused MSCs to the injured site is critical for therapeutic efficacy [9,10]. However, it is known that when MSCs are injected IV, the vast majority of cells become trapped in the pulmonary microvasculature, in what is known as the pulmonary first-past effect [11,12,13,14,15]. To avoid this intrinsic limitation, many groups have opted to instead study stem cell-derived extracellular vesicles (EVs), which are cell-free membrane-bound particles that carry a cargo of regenerative molecules [16,17,18,19]. EVs have been shown to avoid the pulmonary first-pass effect and have a therapeutic effect on par with the MSCs from which they are derived [20]. However, unlike MSCs there is little knowledge on how to further optimize delivery of EVs to target organs [21]. Locoregional delivery, via the arterial blood supply, is perhaps one of the most straightforward and clinically translatable strategies for doing so.

Here, we develop a reproducible microsurgical technique to deliver therapeutics directly into each kidney in the mouse, via its arterial blood supply, while ensuring no operative morbidity or mortality. To demonstrate the therapeutic utility of this technique, we applied it to a mouse model of cisplatin-induced acute kidney injury (AKI), in which we performed IA injection of mesenchymal stromal cell (MSC)-derived EVs.

## 2. Materials and Methods

### 2.1. Animal Protocols

All experimental procedures were performed in accordance with guidelines and regulations of the Administrative Panel on Laboratory Animal Care (APLAC) at Stanford University. A total of 50 healthy CD1 female mice (8 weeks old; body weight 28–31 g) were purchased from Charles River Laboratories (Wilmington, Massachusetts, USA) and housed at constant temperature and humidity and provided with normal feeding in 12:12 light–dark cycles. After 1 week of housing in separate cages in the adaptive environment, mice were randomly divided into 3 groups: group 1 had 10 mice while the other 2 groups had 20 mice each. Group 1 mice were designated as the untreated control group which received normal feeding while the other 2 groups were used to establish AKI models by intraperitonially injecting cisplatin (12 mg/kg) on day 0. To compare the protective effect of IA delivered EVs in cisplatin-induced AKI, group 2 animals (internal control group) received IA injection of normal saline and group 3 animals received IA injection of EVs (150 μg EV protein mass/100 g animal body weight) on day 3. Mice were sacrificed at day 12 after cisplatin injection at which point blood, urine, and kidney samples were collected. One kidney was immediately immersed in 10% neutral buffered formalin for histological analysis while the other kidney was immediately frozen in liquid nitrogen for biochemical marker measurement and western blot analysis.

### 2.2. Extracellular Vesicle Isolation, Characterization, and Purification

In our study we isolated and purified EVs from cultured human bone-marrow derived MSCs (ATCC, Manassas, VA, USA) from three different human donors as previously described [22]. In brief, bone marrow-derived MSCs were cultured in a medium containing 20% fetal bovine serum (FBS) and 100 U/mL penicillin and streptomycin (Thermo Fisher Scientific, Fremont, CA, USA) at 37 °C with 5% CO_2_ until passage 3. The passage 3 cells were cultured for 5 days until they reached 80%–90% confluency and then incubated in serum free Dulbecco’s Modified Eagle Medium (DMEM) overnight to ensure that EVs were originating from the cells and not the serum. Conditioned media was collected, followed by centrifugation at 5000× *g* for 10 min at room temperature to remove cellular debris. The obtained supernatant was ultracentrifuged at 17,000× *g* for 20 min and the EVs were isolated from the supernatant obtained. An anion exchange resin (Q Sepharose Fast Flow, GE Healthcare, IL, USA), which was first balanced with 50 mM NaCl in 50 mM phosphate buffer and then washed with 100 mM NaCl in 50 mM phosphate buffer and later rinsed with 500 mM NaCl in 50 mM phosphate buffer, was used to suspend conditioned medium. We used nanoparticle tracking analysis and transmission electron microscopy (TEM) to measure EV size and number, which ranged from 20 to 180 nm with a mean of 113 nm and a standard deviation of 24 nm (Appendix A). We measured the protein concentration of the EVs using the Pierce bicinchoninic (BCA) Protein Assay Kit (Sigma-Aldrich, St. Louis, MO, USA). The samples were first diluted with radioimmunoprecipitation assay buffer (RIPA) 1:1 and later sonicated in an ice bath for 3 min, and the BCA assay was completed following manufacturer’s instructions. EV surface markers CD9, CD63, and TSG101 were confirmed positive by Western blot analysis (Appendix A).

### 2.3. Intra-Arterial Injection Technique

Our technique for IA injection into the kidneys was optimized in an initial cohort of mice (*n* = 20) in which we established the vascular anatomy relevant to the kidneys and the optimal technique for therapeutic injection to both kidneys. This included examining different injection and suture sites as well as techniques to minimize dissection and operative time while ensuring adequate anatomical exposure. All surgeries were terminal and 100 μL of tattoo dye containing a 1:1 ratio of Solvent Green 3 (dye content 95%) and 1 mg/mL polymethine dye (I2633, Sigma-Aldrich, St. Louis, MO, USA) was used as a surrogate marker to determine the distribution of any therapeutic solution. For survival studies, we used our optimized technique and then administered 100 μL of normal saline, 50 μL to each kidney. Serum creatinine (SCr) and blood urea nitrogen (BUN) levels in mice were measured at baseline and 24 h after intra-arterial injection of saline (Table 1).

Following isoflurane anesthesia, each animal was placed supine and its abdominal wall shaved. The animal was then prepped and draped in the usual sterile fashion and carefully opened with a midline incision, with the intestines carefully displaced cranially to expose each kidney. Both the inferior vena cava (IVC) and aorta were then visualized, with the latter covered by intra-abdominal adipose tissue. 

*Arterial exposure*: The adipose tissue surrounding the aorta was then carefully dissected to reveal the left renal artery, right renal artery, superior mesenteric artery (SMA), and celiac trunk (CT). We found that the safest plane of dissection is to start at the origin of the left renal artery (identified postero-inferiorly to the more prominent left renal vein) and then progress 1 cm superiorly along the length of the aorta to the origin of the right renal artery (which courses posteriorly around the IVC). This provides enough room for ligation of the aorta while also ensuring enough space to cannulate the aorta between the renal arteries. In our experience, we found that the location of the right renal artery is superior to the left renal artery; this is in contrast to humans where the right-sided liver positions the right renal artery slightly inferior to the left renal artery (Figure 1A). Compared to the perpendicularly-angled origin of the right renal artery, the origin of the left renal artery is angled more caudally. Furthermore, we also found in the mouse that the SMA origin is located immediately superior to the right renal artery. Immediately superior to the SMA origin, the CT origin can be found. 

*Vascular preparation to isolate delivery to the kidneys*: A total of three 4-0 nylon sutures (Ethicon) were used to temporarily ligate and isolate vessels at the following sites: two ties were used as a preventive measurement in case of bleeding, 1 cm distal to the left renal artery (i.e., for distal aorta ligation), one tie around the aorta between the SMA and CT (i.e., for proximal aorta ligation), and one tie around the SMA origin (i.e., for SMA ligation). Given that ligation of any vessel risks ischemic damage to the organ(s) which it supplies, our goal was to minimize both the number of vessels ligated and the time of any ligation. Although ligation of the aorta immediately superior and inferior to the renal arteries would be ideal, this was technically not possible due to the anatomical proximity of the SMA and CT to the origin of the right renal artery (Figure 1B). However, with very careful dissection and mobilization of the posterior aorta, we were able to create enough room around the aorta between the CT and SMA to place a proximal suture. A distal suture was then placed around the aorta below the level of the left renal artery and another suture, around the SMA (Figure 1B). None of the looped sutures were tied until immediately before the start of the injection; following sufficient repetition and practice we were able to limit any ischemia time (i.e., from suture ligation to un-ligation) to under 3 min. Once ready for injection, the looped sutures were then ligated in the following order: proximal aorta, SMA, and then distal aorta.

*Injection technique*: We determined that the optimal site of cannulation was immediately distal to the origin of the left renal artery, with the needle entering the vessel with a modest cranial angulation (Figure 1B). Individual cannulation of the renal arteries was not possible due to their very small diameter. In our experience, we found that the largest bore needle which could be safely inserted into the aorta and then removed with hemorrhage control post-injection was a 34-gauge needle. Hence, a 34-gauge needle was then inserted into the aorta with gentle back-tension applied on the distal suture to help keep it in a perfect line to facilitate safe cannulation. Due to the proximity and downward-angle of the left renal artery, our initial studies showed that following injection of tattoo dye into the aorta, first the left kidney was completely stained with almost no staining of the right kidney. This was due to the longer distance from the needle tip to right renal artery, which was also more perpendicularly-angled compared to the left renal artery (Figure 1A). Hence, we applied a metal clamp temporarily on the left renal artery for the first part of the injection to direct any injected solution preferentially into the right renal artery. After 50% of the solution was injected, the metal clamp was then removed and the reminder of the solution was injected which now preferentially flowed into the left kidney. At the end of the injection, the needle was removed and hemostasis was achieved following light pressure over the injection site for 1 min with a Q-tip. The sutures were then unligated in the following order: distal aorta, SMA, and proximal aorta (i.e., distal to proximal) to prevent any barotrauma to the kidneys from the sudden inflow of blood. After confirming hemostasis, all intra-abdominal contents were returned to their original position and the abdominal wall closed using surgical staples. Animals were kept under warm light for 2 h after surgery until fully awake.

### 2.4. Measurements and Analysis of Kidney Function

Urine and blood samples were collected on day 12, at sacrifice. Blood was recovered by heart puncture in heparin-coated capillary blood collection tubes (Terumo, Tokyo, Japan), and it was centrifuged at 3000× *g* for 10 min at 4 °C for measuring blood urea nitrogen (BUN), creatinine (SCr), neutrophil gelatinase-associated lipocalin (NGAL), tumor necrosis factor alpha (TNF-α) and interleukin 6 (IL-6) in plasma. The centrifugation step was repeated twice to minimize platelet contamination, and the clear plasma fraction was stored at −80 °C. The levels of BUN concentrations were measured using the QuantiChrom Urea Assay Kit (DIUR-500, BioAssay Systems, Hayward, California, USA) and creatinine concentrations were measured using an enzyme-linked immunosorbent assay (ELISA; Stanbio, TX, USA). The levels of IL-6 and TNF-α were measured by ELISA kit (R&D Systems, Minneapolis, MN, USA) according to manufacturer’s instructions. Serum NGAL was measured using a NGAL Quantikine ELISA Kit (R&D Systems, USA). Urine samples were collected from both untreated control and both the treatment groups and evaluated for kidney injury molecule-1 (KIM-1), TIMP metallopeptidase inhibitor 1 (TIMP-1) and NGAL. The ELISA kit for KIM-1 and TIMP-1 were purchased from Cell Signaling USA and samples were analyzed as per the manufacturer’s instructions and results were measured and calculated using ELISA reader at 450 nm (Bio-Rad, Irvine, CA, USA).

### 2.5. Western Blotting

After sonication of the already sliced kidney tissues, cells were lysed using cell lysis buffer solution containing 50 mM Tris, pH 7.5, 0.3 M NaCl, 0.5% Triton X-100, 0.1% sodium azide with a mixture of protease inhibitor (Roche, Santa Clara, USA) in 100 μL of buffer for 20 min at 4 °C, then centrifuged at 18,500× *g* for 15 min. The pellet was discarded and the supernatant was kept for further analysis. Protein concentration was quantified using Micro-BCA (Thermo Scientific, USA) for both cell lysates and vesicle preparation in the presence of 0.2% SDS. Equivalent micrograms of proteins, in the case of tissues were homogenized in buffer 150 mM NaCl, 1% Triton X-100, 0.5% deoxycholate, 0.1% SDS, 50 mM Tris, and 1 mM EDTA. Proteins were separated by SDS-PAGE and transferred onto a nitrocellulose membrane. Membranes were then blocked with non-fat milk (5%) for 1 h and probed with primary antibodies at 4 °C overnight. Primary antibodies used in this study were as follows: AMPK (1:1000, sc-39861, Santa Cruz Biotechnologies, USA), phosphorylated AMPK (p-AMPK; 1:500, #2793, Cell Signaling, USA), phosphorylated ERK (pERK; 1:1000, sc-377400, Santa Cruz Biotechnologies, USA), FGF2 (1:1000, sc-365106, Santa Cruz Biotechnologies, USA), Ki67 (1:1000, sc-23900, Santa Cruz Biotechnologies, USA), FGF23 (1:1000, sc-39487, Santa Cruz Biotechnologies, USA), NF-KB (1:500, sc-8414, Santa Cruz Biotechnologies, USA). β-actin (sc-47778, Santa Cruz Biotechnology, Texas, USA) was used as an internal control. The membranes were then washed with PBS/0.01% Tween and incubated with anti-rabbit secondary antibody to IgG coupled to horseradish peroxidase at room temperature for 2 h (1:5000) (sc-2748, Santa Cruz Biotechnologies). Quantification was measured with the aid of chemiluminescence Western blotting substrate (Roche) which enhanced chemiluminescence, and a ChemiDoc imager (Bio-Rad) was used to measure the intensity of the signals. 

### 2.6. Immunohistochemistry and Histology

The kidney tissues were fixed in 4% (v/v) paraformaldehyde in PBS for 24 h and later on imbedded in paraffin. Tissues were then processed and sectioned (6 μm) for immunostaining and histological staining. The sections were de-paraffinized and dehydrated by immersing them in graded ethanol concentrations for 1 h. Then antigens were retrieved using 10 mM sodium citrate buffer (pH 8.5) at 80 °C for 30 min. The tissue slices were blocked and permeabilized with PBS containing 0.2% Triton X-100 (Sigma-Aldrich, St. Louis, MO, USA) and 2% bovine serum for 30 min at room temperature. The sections were incubated overnight at 4 °C with the primary rabbit anti-human antibody to TNF-α (1:200, #11948, Cell Signaling, Danvers, MA, USA), NF-KB (1:200, #6956, Cell Signaling, Danvers, MA, USA), FGF2 (1:200, #5414, Cell Signaling, Danvers, MA, USA), or Ki67 (1:200; MA5-14520, Thermo Scientific, Fremont, CA, USA). Following incubation with Alexa 594-conjugated anti-rabbit antibody to immunoglobulin G (1:1000, A-1102, Thermo Fisher Scientific, Fremont, CA, USA) and horseradish peroxidase (HRP) at room temperature for 2 h, 3,3′-diaminobenzidine (DAB) substrate (1 μg/mL, Cell Signaling, Danvers, MA, USA) was added to the sections and let stand for 30 min. Sections were stained using hematoxylin and eosin (H&E) stain (Thermo Scientific, Fremont, CA, USA) and observed under an optical microscope (Nikon, San Diego, CA, USA). 

To determine the tubular casts score, hematoxylin and eosin stained preparations were evaluated under a light microscope. Tubular casts were assessed in non-overlapping fields (up to 20 for each section) using 40 objective images, in a single-blind fashion. Scores were assigned by calculating the percentage of tubules positive for cast formation. Kidneys showing no injury were marked 0. Kidneys exhibiting minimal (<10%), mild (10%–25%), moderate (26%–50%), extensive (51%–75%), and severe (≥75%) injuries were assigned scores of 1, 2, 3, 4, and 5, respectively.

### 2.7. Real-Time PCR

The total RNA was extracted from kidney homogenized tissue using Trizol reagent (Sigma-Aldrich, St. Louis, Missouri, USA). After digestion with DNase I, 2 µg RNA was reverse transcribed into cDNA as per the instructions in the Applied Biosystems reverse transcriptase kit (Applied Biosystems, CA, USA). The quality of cDNA was assessed by the ratio of the absorbance at 260 nm and 280 nm using an Agilent 2100 Bioanalyzer (Agilent Bioanalyzer, CA, USA). cDNA was then amplified by PCR in an iCycler Thermal Cycler (Bio-Rad, CA, USA) with SYBR Green (Applied Biosystems, CA, USA) and specific primers for KIM-1 (PA5-79345, Applied Biosystems, CA, USA), TIMP-1 (MS-608-PABX, Applied Biosystems, CA, USA), NAGL (PA5-79591, Applied Biosystems, CA, USA), NF-KB (Applied Biosystems, PA5- 27340, CA, USA), and GAPDH (Applied Biosystems, MA5-15738, CA, USA) were used. The relative expression of mRNAs was calculated by the 2^−ΔΔCt^ method and normalized to glyceraldehyde 3-phosphate dehydrogenase (GAPDH).

### 2.8. Statistical Analysis

The measured data was expressed as mean ± SD. Comparison between two or more groups is done by one-way ANOVA with Tukey’s multiple comparison test. Graph Pad Prism software (GraphPad, San Diego, CA, USA), was used for all analysis, and the values of *p* < 0.05 were considered statistically significant.

## 3. Results

### 3.1. IA Injection of Dye Selectively and Uniformly Labels the Kidneys

Our procedures for arterial exposure, vascular preparation, and injection are detailed in depth in the Methods section. Briefly, following exposure of the aorta, three sites were ligated with sutures: (1) the proximal aorta between the origins of the celiac trunk (CT) and superior mesenteric artery (SMA), (2) the SMA, and (3) the distal aorta (Figure 1B,C). A metal clamp was placed temporarily on the left renal artery to first allow delivery to the right renal artery; after injection of 50% of the volume, the clamp was removed, which results in preferential delivery to the left kidney due to natural flow.

Injection of the tattoo dye into the distal aorta, without application of the described ligation technique, resulted in non-targeted staining of the liver, lungs, and preferential staining of the left kidney due to the proximity and downward angle of the left renal artery origin (Figure 1D). However, injection of tattoo dye with our outlined technique resulted in targeted homogenous staining of both kidneys, without evidence of any systemic non-target staining (Figure 1E,F).

In our experience, the entire surgical procedure to deliver therapeutics, via IA injection into both kidneys, can be performed in approximately 35 min with no significant animal morbidity or mortality. Analysis of the serum from animals undergoing this procedure demonstrated no significant biochemical changes suggestive of kidney injury, including serum creatinine (SCr) and blood urea nitrogen (BUN) (Table 1). Any functionally significant iatrogenic kidney damage would have been detectable within this time period, without unnecessary sacrifice of animals for histologic biopsy. In addition, there was no mortality up to 12 days after the procedure in any of our experimental animals (Figure 2C).

### 3.2. Intra-Arterial EV Treatment Restores Kidney Function Following AKI

Following the induction of AKI with cisplatin and IA injection of saline, there was significant reduction in survival (100% vs. 50%), body weight (23.8 ± 1.6 g vs. 19.2 ± 3.4 g, *p* < 0.05), and kidney weight (0.21 ± 0.02 g vs. 0.15 ± 0.02 g, *p* < 0.05), with grossly visible atrophy and discoloration (Figure 2B,C) compared to untreated control animals. Compared to the AKI + IA saline group, animals treated with IA EVs showed improved survival (50% vs. 80%), body weight (19.2 ± 3.4 g vs. 20.9 ± 2.1 g, *p* > 0.05), and kidney weight (0.15 ± 0.02 g vs. 0.19 ± 0.02 g, *p* < 0.05), as well as grossly-visible improvements in the size and color of the harvested kidneys.

Compared to untreated controls, animals in the AKI + IA saline group showed significantly elevated BUN (22.19 ± 6.22 vs. 151.32 ± 13.59 mg/dL, *p* < 0.05), SCr (0.37 ± 0.05 vs. 3.95 ± 0.43 mg/dL, *p* < 0.05), and serum NGAL (1.05 ± 0.21 vs. 4.26 ± 1.36 mg/mL, *p* < 0.05) (Figure 2D). Compared with the AKI + IA saline group, those treated with IA EVs had significantly reduced BUN (151.32 ± 13.59 vs. 48.28 ± 5.87 mg/dL, *p* < 0.05) and SCr (3.95 ± 0.43 vs. 1.67 ± 0.30 mg/dL, *p* < 0.05) as well as decreased NGAL which did not reach statistical significance (4.26 ± 1.36 vs. 3.47 ± 0.94 mg/mL, *p* > 0.05).

Similar results were seen after measuring mRNA levels of injury molecules in the kidney tissue, KIM-1, TIMP-1, and NGAL. Compared to untreated controls, animals in the AKI + IA saline group showed significant upregulation in KIM-1 mRNA (44.2 ± 6.4 vs. 176.2 ± 9.8 relative expression, *p* < 0.05), TIMP-1 (4.6 ± 1.3 vs. 26.5 ± 2.2 relative expression, *p* < 0.05), and NGAL (6.7 ± 0.9 vs. 37.0 ± 2.5 relative expression, *p* < 0.05). Compared to the AKI + IA saline animals, those treated with IA EVs had significant downregulation of KIM-1 mRNA (176.2 ± 9.8 vs. 55.0 ± 11.3 relative expression, *p* < 0.05) and NGAL mRNA (37.1 ± 2.5 vs. 12.2 ± 2.7 relative expression, *p* < 0.05), as well as in TIMP-1 mRNA which did not reach statistical significance (26.5 ± 2.2 vs. 19.6 ± 3.0 relative expression, *p* > 0.05) (Figure 2E).

These trends were also recapitulated by measuring urine concentrations of kidney injury molecules. Compared to untreated controls, animals in the AKI + IA saline group had significant increases in urine KIM-1 (3.2 ± 1.0 vs. 21.8 ± 3.1 mg/dL, *p* < 0.05), TIMP-1 (0.8 ± 0.3 vs. 4.7 ± 1.7 mg/dL, *p* < 0.05), and NGAL (3.7 ± 1.1 vs. 20.0 ± 2.8 mg/dL, *p* < 0.05) (Figure 2F). Compared to the AKI + IA animals, treatment with IA EVs significantly decreased urine KIM-1 (21.8 ± 3.1 vs. 1.6 ± 1.1 mg/dL, *p* < 0.05), decreased urine NGAL though not reaching statistical significance (20.0 ± 2.8 vs. 13.7 ± 4.5 mg/dL, *p* > 0.05), and had no effect on urine TIMP-1 (4.7 ± 1.7 vs. 6.1 ± 0.9 mg/dL, *p* > 0.05).

### 3.3. Intra-Arterial EV Treatment Attenuates Kidney Inflammation

As inflammation is one of the major mechanisms by which AKI causes tissue damage, we analyzed the ability of IA EVs to exert anti-inflammatory effects. Histological examination of kidney tissue from the AKI + IA saline group revealed extensive tubular cast formation compared to untreated control group (0.06 ± 0.03 vs. 3.25 ± 0.52 score, *p* < 0.05), indicative of kidney injury (Figure 3A). Mice treated with AKI + IA EVs, however, showed significant decrease in cast formation compared to the AKI + IA saline group (3.25 ± 0.52 vs. 1.64 ± 0.47 score, *p* < 0.05). Immunohistochemical staining of the kidney showed that compared to untreated controls, those in the AKI + IA saline group had significant elevation of the proinflammatory cytokine TNF-α and its downstream effector NF-κB (Figure 3A). Treatment with IA EVs decreased the signal of both these markers. Compared to untreated controls, serum levels of proinflammatory cytokines were significantly upregulated in the AKI + IA saline group, including TNF-α (469.2 ± 68.8 vs. 1597.4 ± 291.4 pg/mL, *p* < 0.05), and IL-6 (414.9 ± 35.1 vs. 1345.1 ± 244.2 pg/mL, *p* < 0.05) (Figure 3B). Compared to the AKI + IA saline group, those treated with IA EVs had significant downregulation of both TNF-α (1597.4 ± 291.4 vs. 698.7 ± 125.3 pg/mL, *p* < 0.05) and IL-6 (1345.1 ± 244.2 vs. 694.5 ± 244.2 pg/mL, *p* < 0.05).

Similar trends were seen in protein and mRNA expression of NF-κB, as measured in kidney lysate by Western blot and qRT-PCR, respectively (Figure 3C,D). Compared to the untreated controls, the AKI + IA saline group had upregulation of NF-κB protein (1.5 ± 0.4 vs. 3.6 ± 0.6 relative expression, *p* > 0.05) and NF-κB mRNA (0.4 ± 0.1 vs. 5.9 ± 1.9 relative expression, *p* > 0.05). Compared to the AKI + IA saline group, those treated with IA EVs had downregulation of NF-κB protein (3.6 ± 0.6 vs. 1.5 ± 0.3 relative expression, *p* > 0.05) and NF-κB mRNA (5.9 ± 1.9 vs. 2.6 ± 0.6 relative expression, *p* > 0.05), though these differences did not reach statistical significance.

### 3.4. Intra-Arterial EV Treatment Promotes Proliferation

Cell proliferation is another major mechanism by which EVs exert their therapeutic effect. Immunohistochemical staining of the kidney showed that compared to untreated controls, those in the AKI + IA saline group had lower expression of proliferation markers FGF2 and Ki67 (9.2 ± 1.9% vs. 3.8 ± 0.8% Ki67+, *p* > 0.05) (Figure 4A). Compared to the AKI + IA saline group, those treated with IA EVs showed significant increase in Ki67+ cells (3.8 ± 0.8% vs. 6.5 ± 0.8% Ki67+, *p* < 0.05).

These results were further corroborated via Western blot on kidney lysates. Compared to untreated controls, animals in the AKI + IA saline group had downregulation of Ki67 (0.6 ± 0.2 vs. 0.2 ± 0.1 relative expression, *p* > 0.05) and FGF2 protein (1.7 ± 0.6 vs. 0.4 ± 0.1 relative expression, *p* > 0.05), as well as upregulation of FGF23, a marker of kidney disease (0.5 ± 0.2 vs. 2.5 ± 0.5 relative expression, *p* < 0.05) (Figure 4B). Compared to AKI + IA saline group, those treated with IA EVs showed upregulation of Ki67 (0.2 ± 0.1 vs. 0.4 ± 0.1 relative expression, *p* > 0.05) and FGF2 (0.4 ± 0.1 vs. 1.9 ± 0.3 relative expression, *p* < 0.05), as well as downregulation of FGF23 (2.1 ± 0.5 vs. 0.5 ± 0.1 relative expression, *p* < 0.05).

To investigate the underlying signaling pathways responsible for these changes in proliferation, we conducted Western blot analysis of AMPK, a central regulator of cellular metabolism and growth, and ERK, a classical proliferation pathway. Compared to untreated controls, those in the AKI + IA saline group had significant downregulation of both pAMPK (3.4 ± 0.4 vs. 0.3 ± 0.1 relative expression, *p* < 0.05) and pERK (2.9 ± 0.2 vs. 1.3 ± 0.2 relative expression, *p* < 0.05) (Figure 4C). Compared to the AKI + IA group, those treated with IA EVs had significant upregulation of pAMPK (0.3 ± 0.1 vs. 2.6 ± 0.4 relative expression, *p* < 0.05) and upregulation of pERK that did not reach statistical significance (1.3 ± 0.2 vs. 1.6 ± 0.2 relative expression, *p* > 0.05).

## 4. Discussion

In this study, we have developed a microsurgical technique for the injection of therapeutics such as EVs into the mouse kidney, directly via its arterial blood supply. Our method is reproducible, while ensuring no operative morbidity or mortality. Injection of tattoo dye using our method resulted in homogeneous staining of both kidneys, with almost no staining of off-target organs. We demonstrated the utility of our method by applying it to a mouse model of cisplatin-induced AKI. Intra-arterially delivered EVs successfully restored physiological measures of kidney function, reduced histological and molecular markers of injury, attenuated local inflammation, and restored proliferative and regenerative signaling, demonstrating its potent regenerative potential. 

Intra-arterial delivery of therapeutics to the kidney can be desirable for several reasons. These include minimizing systemic side effects while simultaneously increasing therapeutic efficacy; this can be attributed to increasing the delivery of the therapeutic cargo to the target site and avoiding of any potential first-pass metabolism for drugs or sequestration for cellular therapies. The development of IA therapeutics for kidney disease, however, relies on the availability of animal models and feasibility of surgical techniques. Previous studies that injected MSCs directly into the renal artery have used larger animal models, including cats [23], pigs [24], rats [25], and sheep [26]. Indeed, a meta-analysis of 21 studies applying MSC therapy to animal models of renal failure has suggested that IA delivery of MSCs has greater therapeutic effect compared to intravenous delivery [27]; it should be noted though that the studies used different animal and disease models, and none of them did a direct comparison between delivery routes. Regardless, these larger animals are often not readily accessible to researchers, or do not have as well-established models of kidney disease. Mice are by far the most comprehensive species in regards to models of renal pathology, with established models of acute kidney injury (AKI), chronic kidney disease (CKD), and kidney cancer, via nephrotoxic, genetic, autoimmune, metabolic, and ischemic etiologies [28,29]. However, to our knowledge, there have not been any previous studies delivering therapeutics directly into the renal artery of mice [27], in part due to the technically challenging nature of injecting small vessels. Our technique thus paves the way for preclinical exploration of IA therapeutics for a wide range of renal pathologies.

As a demonstration of our technique, we applied it to a mouse model of cisplatin-induced AKI, treated with MSC-derived EVs. EVs carry a cargo of therapeutic molecules that suppress inflammation and apoptosis and promote regeneration, and have shown promising preclinical results for treating AKI [16]. However, whereas MSCs are known to home to sites of injury [30,31], EV biodistribution in the context of injury is far less-studied [32]. In the brain, in vivo neuroimaging of gold-labeled EVs has shown that EVs home to areas of brain pathologies in mice, but distribute diffusely in uninjured brains [33,34]. One study has shown that human umbilical vein endothelial cell (HUVEC)-derived EVs home to ischemic kidneys, but not uninjured kidneys, for at least 4 h post-injection, an effect likely mediated by CXCR4/SDF-1α interactions [35]. Even less studied are methods to further optimize EV homing to specific target tissues. While various surface engineering, genetic, and adjuvant approaches may enhance EV homing [21,32,36], these approaches are often resource-costly; locoregional delivery may be one of the most direct ways to optimize EV therapy. Our IA delivery of EVs restored physiological markers of kidney function (BUN, creatinine), reduced histological and molecular markers of kidney injury (KIM-1, TIMP-1, NGAL, FGF23) [37], attenuated inflammation (TNF-α, IL-6), and spurred cellular proliferation (Ki67, FGF2). These effects may have been mediated through AMPK, which has a documented role in kidney regeneration by modulating energy metabolism, inflammation, and stress [38], as well as MAPK/ERK, a classical pathway promoting cellular proliferation [39]. Taken together, these results offer proof of principle of the feasibility and efficacy of IA delivery of therapeutics into the kidney. Future studies will aim to compare the effect of different MSC-based therapies (parent MSCs vs. MSC-derived EVs) through different administration routes (intravenous, intra-arterial, or intraperitoneal), including tracking their biodistribution of degradation over time. Such knowledge is critical for the effective translation of preclinical studies into feasible and effective therapeutics.

Finally, a point on the clinical-translational aspect of our technique deserves brief discussion here. Obviously, our microsurgical technique is quite invasive to the mouse, requiring opening of the peritoneal cavity and exposure of the abdominal aorta. However, establishing access to the renal artery in humans is relatively simple given modern interventional radiology techniques. Percutaneous access to the renal artery is easily achieved through the femoral, brachial, or radial arteries, and such techniques have long been in use for the treatment of conditions like renal artery stenosis [40]. EVs may outperform cellular therapies in this application, as their smaller size reduces the risk of embolic events and can be injected through catheters with less damage and shear stress. Thus, our technique is one that allows rapid translation to clinical trials given the proper indications.

In summary, we have developed the first known microsurgical technique for the direct delivery of therapeutics into the mouse kidney via its arterial blood supply. This technique should now allow researchers to study the effects of direct delivery of different oncological and regenerative therapies into the kidneys within the many well-established mouse models of renal pathology, with rapid clinical translation possible through modern interventional radiology techniques.

## Figures and Tables

**Figure 1 cells-09-00937-f001:**
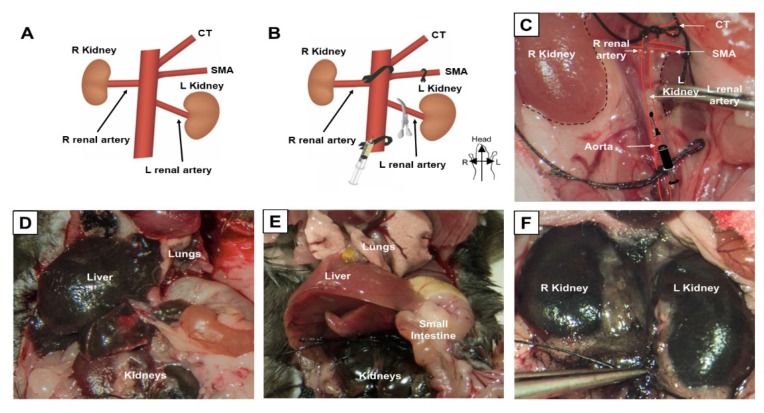
Selective ligation allows for selective kidney labeling. (**A**) Normal anatomy of the mouse abdominal aorta, showing origin of the celiac trunk (CT), superior mesenteric artery (SMA), renal arteries, and kidneys. (**B**) Suture ligation sites, including the proximal aorta between the origins of the CT and SMA, the SMA, and the distal aorta. A metal clip is placed temporarily on the left renal artery to allow delivery of therapeutics first to the right renal artery. (**C**) Selective ligation and clip placement shown inside the mouse abdomen. (**D**) Result of tattoo dye injection into the distal aorta without described ligation technique. (**E**,**F**) Result of tattoo dye injection with our ligation and clipping technique.

**Figure 2 cells-09-00937-f002:**
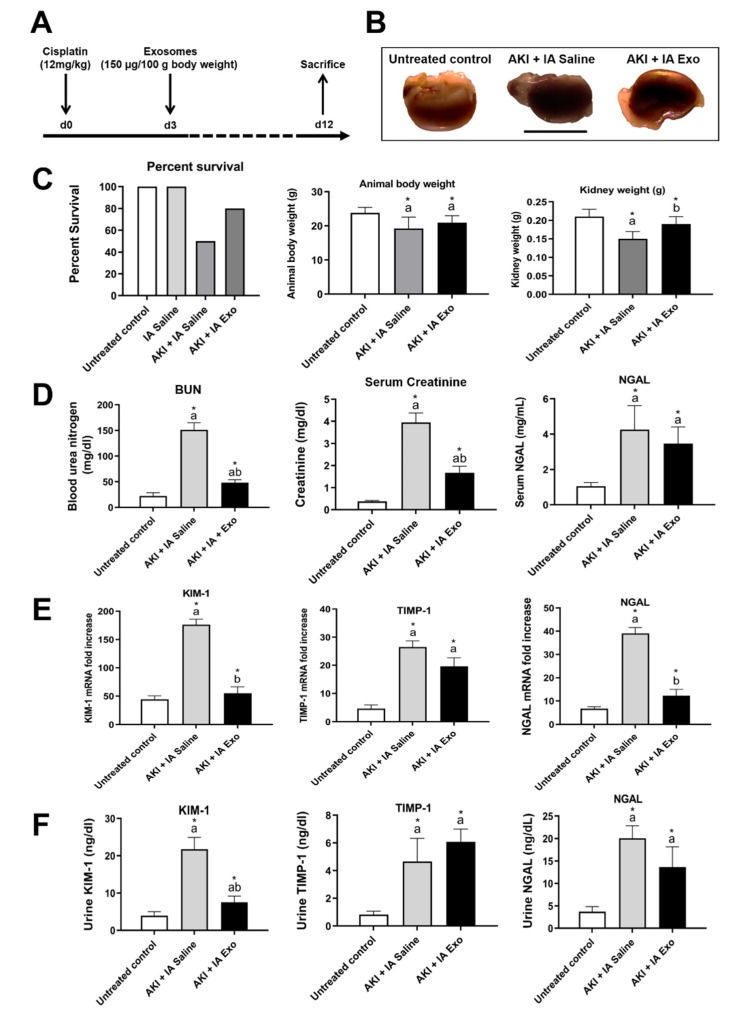
Physiological and molecular markers of kidney injury. (**A**) Study protocol. (**B**) Gross appearance of kidney. (**C**) Percent survival, animal body weight, and kidney weight. (**D**) Serum concentration of blood urea nitrogen (BUN), creatinine, and NGAL, as measured by serum ELISA. (**E**) mRNA expression of KIM-1, TIMP-1, and NGAL in kidney lysates, as measured by qRT-PCR. (**F**) Urine concentrations of KIM-1, TIMP-1, and NGAL, as measured by urine ELISA. Measurements were taken at day 12. Each group has *n* = 8 mice, except for survival data for which *n* = 10 for untreated controls and *n* = 20 mice for other groups. Significant difference ^a^
*p* < 0.05: relative to untreated control group; ^b^
*p* < 0.05: relative to AKI + IA saline group.

**Figure 3 cells-09-00937-f003:**
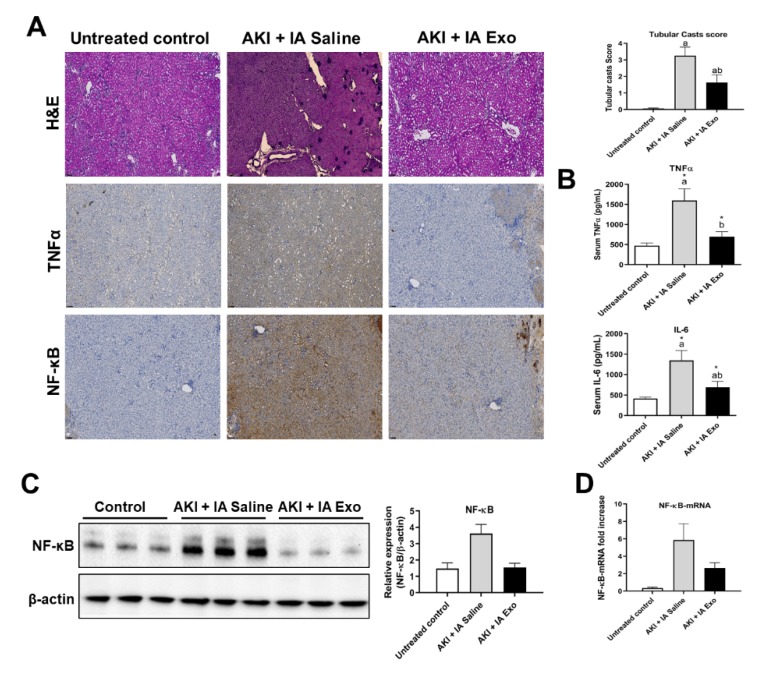
Inflammatory cytokines. (**A**) Hematoxylin and eosin (H&E) staining and immunohistochemistry (IHC) of kidney tissue, stained for TNF-α and NF-κB, and quantification of tubular casts. (**B**) Serum concentrations of TNF-α and IL-6, as measured by serum ELISA. (**C**) Western blot and quantification of NF-κB and β-actin from kidney lysate. (**D**) mRNA expression of NF-κB, as measured by qRT-PCR. Measurements were taken at day 12. Each group has *n* = 5 mice for TNF-α and IL-6 measurements and *n* = 3 mice for NF-κB measurements. Significant difference ^a^
*p* < 0.05: relative to untreated control group; ^b^
*p* < 0.05: relative to AKI + IA saline group.

**Figure 4 cells-09-00937-f004:**
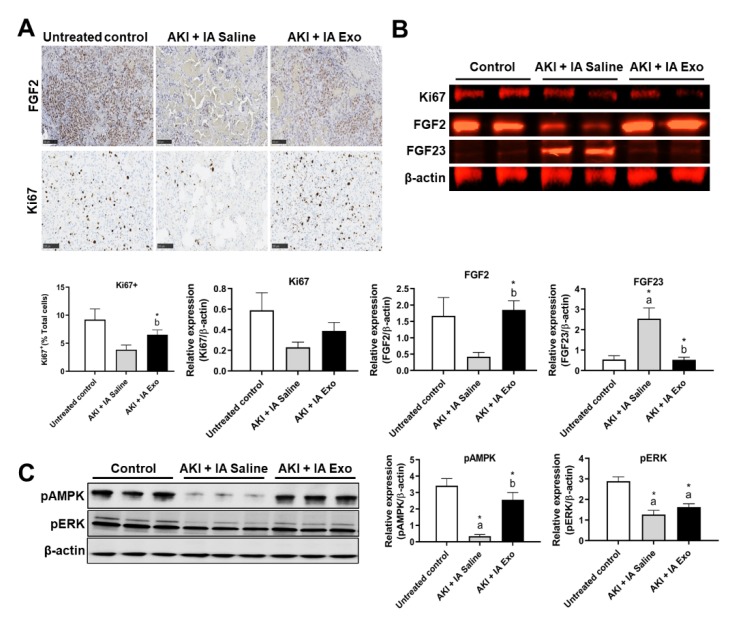
Proliferation and regeneration markers. (**A**) Immunohistochemistry (IHC) of FGF2 and Ki67 in kidney tissue, and quantification of Ki67+ cells. (**B**) Western blot and quantification of Ki67, FGF2, FGF23, and β-actin from kidney lysate. (**C**) Western blot and quantification of pAMPK, pERK, and β-actin from kidney lysate. Measurements were taken at day 12. Each group has *n* = 3 pooled mice. Significant difference ^a^
*p* < 0.05: relative to untreated control group; ^b^
*p* < 0.05: relative to AKI + IA saline group.

**Table 1 cells-09-00937-t001:** Comparative markers of renal function pre and post injection. Serum creatinine (SCr) and blood urea nitrogen (BUN) in mice at baseline and 24 h after intra-arterial injection of saline.

	Reference Ranges	Baseline	Post-Operative (24 h)
**SCr (mg/dL)**	0.1–1.1	0.2 ± 0.1	0.1 ± 0.1
**BUN (mg/dL)**	20.3–24.7	26.0 ± 2.3	28.0 ± 6.4

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
