# Peer review of "A Novel Approach to Deliver Therapeutic Extracellular Vesicles Directly into the Mouse Kidney via Its Arterial Blood Supply"

_cells, 2020, doi:10.3390/cells9040937_

Round 1

Reviewer 1 Report

The authors have established a technique for the direct delivery of therapeutics into the mouse kidney via its arterial blood supply. Using this technique, they also showed that the delivered exosomes derived from bone marrow mesenchymal stroma cells restored cisplatin-induced renal injury. These are very interesting observations. I have some minor comments.

Minor comments

  1. Figure 2C: How many animals were used in the IA saline group?
  2. What did tattoo fluid dye? What is the index of dyeing ability with tattoo liquid?
  3. At line 84: How did the authors measure the weight of exosomes?
  4. There are discrepancies between the values shown in the results at line 287 and 290, 294 and 296, 247 and 350. Please correct them.
  5. At line 342-343: Why don't you show numerical data for FGF2 experiments?
  6. I recommend you to make a space before the unit, and before and after the symbol, such as micro g, micro m, nm, ±, and so on.
  7. Please indicate the catalog number of all antibodies used in the paper?

Author Response

Reviewer 1

The authors have established a technique for the direct delivery of therapeutics into the mouse kidney via its arterial blood supply. Using this technique, they also showed that the delivered exosomes derived from bone marrow mesenchymal stroma cells restored cisplatin-induced renal injury. These are very interesting observations. I have some minor comments.

Minor comments

Figure 2C: How many animals were used in the IA saline group?

For the survival data, each group has n = 20 mice, except for the untreated controls, which had n = 10 mice. These sample sizes are noted in the caption of Figure 2 (lines 320-321).

What did tattoo fluid dye? What is the index of dyeing ability with tattoo liquid?

We thank the reviewer for pointing out this ambiguity in the methods. The tattoo dye used in this study contained 1:1 ratio of Solvent Green 3 (dye content 95%) and 1 mg/mL polymethine dye (Sigma Aldritch, I2633), which is routinely used for visualization of tissues following its administation in experimental animal models. The methods section 2.3 has been updated with this information (lines 114-115).

Based on our observations, injection of tattoo dye into the aorta with no additional ligation resulted in staining of the liver, lungs, and left kidney (Figure 1D). With the ligation technique, however, we achieved homogeneous focal labeling of both kidneys (Figure 1E, F).

At line 84: How did the authors measure the weight of exosomes?

We thank the reviewer for pointing out this ambiguity in the methods. Mice in the IA + EV group were injected with 150μg EV protein mass per 100g animal body weight. The total protein concentrations of EVs were measured using Pierce™ BCA Protein Assay Kit (Sigma Aldrich, USA). First the samples were diluted 1:1 with RIPA and then sonicated on ice bath for 3 min, and the BCA assay was completed following the manufacturer’s instructions. Both sections 2.1 (lines 84-85) and 2.2 (lines 104-107) have been amended accordingly.

There are discrepancies between the values shown in the results at line 287 and 290, 294 and 296, 247 and 350. Please correct them.

We thank the reviewer for catching these inconsistencies. The numerical values reported in the manuscript are all correct, but the order in which we compared the numbers may have caused confusion. We have accordingly reordered the numbers to be more clear as to which numbers correspond to which experimental group.  We apologize for any confusion caused

At line 342-343: Why don't you show numerical data for FGF2 experiments?

We agree with the reviewer that the FGF2 quantification is missing. Unfortunately, due to the COVID-19 pandemic and shelter-in-place order in effect, we do not currently have access to the original IHC slides or additional digital images to perform rigorous quantification of FGF2+ cells. We are, however, confident in our data as the trends are well-recapitulated in the Western blot data, whose quantification is provided (Figure 4B). We apologize for our inability to amend this omission at the time.

I recommend you to make a space before the unit, and before and after the symbol, such as micro g, micro m, nm, ±, and so on.

We agree with the reviewer on unit style conventions and have made all appropriate changes in the text.

Please indicate the catalog number of all antibodies used in the paper?

We agree with the reviewer that catalog numbers for antibodies are important for replicability. All catalog numbers are now included in the Methods section.

Reviewer 2 Report

Ullah et al. reported a novel IA approach for delivering therapeutics to AKI mouse. The therapeutics are MSC exosomes in this study. The major focus is on the novel delivery method, but the authors also show some improvements of AKI model with exosome transplantation. Indeed, finding optimized transplantation strategies in small animals for kidney disease is critical for research. This manuscript is well written, and the experiment is well designed. However, some minor issues need to be revised and some improvements can be made in the revision:

1, I assume this is a mistake: in MM, the author claimed culture MSCs at room temperature, please explain or revise. Moreover, for passage 3 cells, it is not clear how long the author cultured the cells with serum-free DMEM. If it is over night, the cells are not likely to become 80-90 confluency. Please revise the whole cell culture and exosome isolation part.

2, could the author add the size distribution of exosomes based on TEM images? And compare this with the results from NTA.

3, from line 284-291, the notation is wrong, should be Figure 2E. Similarly, line 295 it should be Figure 2F. Please carefully revise this.

4, in discussion, the authors claim that exosomes are not known to home to injury site as MSCs. In some cases, this is not correct (see reference below). Thus, it is very important to know the biodistribution and degradation of exosomes after IA administration. I know it is hard to add animal experiment, but it is worthy to discuss this matter.

https://doi.org/10.1117/12.2287351;

DOI: 10.1021/acsnano.7b04495;

DOI: 10.1021/acs.nanolett.8b04148.

5, it is not clear how the AMPK pathway is involved in all the therapeutic improvements in this study. For example, how the endogenous AMPK to be associated with immunomodulation of the MSC exosomes? How the inflammation is regulated by the metabolic changes in kidney disease especially AKI. It is better to discuss the molecular/physiological details on the MSC/exosome therapy in kidney diseases.

6, There are several grammatical errors and typos in the manuscript.

Author Response

Reviewer 2

Ullah et al. reported a novel IA approach for delivering therapeutics to AKI mouse. The therapeutics are MSC exosomes in this study. The major focus is on the novel delivery method, but the authors also show some improvements of AKI model with exosome transplantation. Indeed, finding optimized transplantation strategies in small animals for kidney disease is critical for research. This manuscript is well written, and the experiment is well designed. However, some minor issues need to be revised and some improvements can be made in the revision:

1. I assume this is a mistake: in MM, the author claimed culture MSCs at room temperature, please explain or revise. Moreover, for passage 3 cells, it is not clear how long the author cultured the cells with serum-free DMEM. If it is over night, the cells are not likely to become 80-90 confluency. Please revise the whole cell culture and exosome isolation part.

We thank the reviewer for catching these oversights in the text. The relevant section has been corrected accordingly (lines 90-98):

“In our study we isolated and purified EVs from cultured human bone-marrow derived MSCs (ATCC, USA), from three different human donors as previously described [22]. In brief, bone marrow-derived MSCs were cultured in a medium containing 20% fetal bovine serum (FBS) and 100 U/mL penicillin and streptomycin (Thermo Fisher Scientific, USA) at 37 °C with 5% CO2 until passage 3. The passage 3 cells were cultured for 5 days until they reached 80%–90% confluency and then incubated in serum free DMEM overnight to ensure that EVs were originating from the cells and not the serum. Conditioned media was collected, followed by centrifugation at 5,000×g for 10min at room temperature to remove cellular debris.”

2. could the author add the size distribution of exosomes based on TEM images? And compare this with the results from NTA.

We thank the reviewer for the suggestion. While we do not have the software to systematically analyze the size distribution of EVs using TEM, we have now manually measured the sizes of EVs from the TEM images using ImageJ. The size distribution obtained from our manual TEM measurements had a mean 100.6nm with SD 21.3nm, which was remarkably similar to the results obtained from NTA (mean 113nm with SD 24nm). We realize that this is not exactly the most rigorous way to quantify the TEM images, but it is the best we can do under the current lock-down circumstances. A representative TEM image with manual measurements is included below (note that the clumps are not measured as these are artefacts), as well as the corresponding size histogram.

[Revision Figures can be found in the attached PDF]

3. from line 284-291, the notation is wrong, should be Figure 2E. Similarly, line 295 it should be Figure 2F. Please carefully revise this.

We thank the reviewer for catching these errors. The figure numbers in the text have now all been carefully revised.

4. in discussion, the authors claim that exosomes are not known to home to injury site as MSCs. In some cases, this is not correct (see reference below). Thus, it is very important to know the biodistribution and degradation of exosomes after IA administration. I know it is hard to add animal experiment, but it is worthy to discuss this matter. https://doi.org/10.1117/12.2287351; DOI: 10.1021/acsnano.7b04495; DOI: 10.1021/acs.nanolett.8b04148.

We thank the reviewer for bringing this literature to our attention. We have added a discussion regarding the homing of EVs within the brain, including the references provided (lines 413-418).

Regarding the biodistribution and degradation of EVs after IA administration, we agree that these are important experiments to perform. Accordingly, our group will be studying fluorescently labelled EVs using the following technique: fluorescent dye-stained EVs will be injected IA and mice will be sacrificed after 24h post-IA injection. All organs will then be isolated to measure the fluorescent signal from any retained EVs using an IVIS Spectrum at a wave-length of 745nm/ 800nm. The radiant effectiveness of the tissue samples is measured using a Living Image Software with the relative mean fluorescent signal from the tissue sample calculated by subtracting the mean fluorescent signal from control mice (no EV injection). Unfortunately, we cannot currently proceed with these experiments for this manuscript due to the COVID-19 pandemic, but we have emphasized the importance of these future experiments in the text (lines 430-431).

Reviewer 3 Report

The authors described a new microsurgical technique for the direct delivery of therapeutics into the mouse kidney via its arterial blood supply. This therapy consists in mesenchymal stromal cell-derived exosomes. Intra-arterially delivered exosomes successfully restored physiological measures of kidney function, reduced histological and molecular markers of injury, attenuated local inflammation, and restored and regenerative signaling, demonstrating its potent regenerative potential. 

Concerns:

1.What is the source of the mesenchymal stromal cells used in this work?

2.The authors do not mention the volumen and the brand of the antibodies used to characterised the exosomes in the main text.

The ISEV consensus recommendation on nomenclature is to use “extracellular vesicle” as the “generic term for particles naturally released from the cell that are delimited by a lipid bilayer and
cannot replicate” and to modify “EV” based on clear,measurable characteristics such as cell of origin, molecular markers, size, density, function, etc. (Théry C, Witwer KW, Aikawa E, et al. Minimal information for studies of extracellular vesicles 2018 (MISEV2018): a
position statement of the International Society for Extracellular Vesicles and update of the MISEV2014 guidelines. J Extracell Vesicles. 2018;7(1):1535750.

3.Why do the authors keep writing about exosomes instead extracellular vesicles?

4.The figure 4B seems very satured. Could the authors show a new one?

5.Page 10, line 370. The authors write morality. Is it correct? I think that the authors wanted to write mortality.

Author Response

Reviewer 3

The authors described a new microsurgical technique for the direct delivery of therapeutics into the mouse kidney via its arterial blood supply. This therapy consists in mesenchymal stromal cell-derived exosomes. Intra-arterially delivered exosomes successfully restored physiological measures of kidney function, reduced histological and molecular markers of injury, attenuated local inflammation, and restored and regenerative signaling, demonstrating its potent regenerative potential.

Concerns:

1.What is the source of the mesenchymal stromal cells used in this work?

The MSCs used in this study were human bone marrow-derived MSCs (ATCC, USA), from three different human donors. Extracellular vesicles were purified from these MSCs as described in Kim et al., PNAS (2016). These details can be found in section 2.2 of the manuscript, which has now been amended (lines 90-91).

2.The authors do not mention the volumen and the brand of the antibodies used to characterised the exosomes in the main text.

We agree with the reviewer that catalog numbers and dilutions for antibodies are important for replicability. All catalog numbers and dilutions are now included in the Methods section.

3. The ISEV consensus recommendation on nomenclature is to use “extracellular vesicle” as the “generic term for particles naturally released from the cell that are delimited by a lipid bilayer and cannot replicate” and to modify “EV” based on clear,measurable characteristics such as cell of origin, molecular markers, size, density, function, etc. (Théry C, Witwer KW, Aikawa E, et al. Minimal information for studies of extracellular vesicles 2018 (MISEV2018): a position statement of the International Society for Extracellular Vesicles and update of the MISEV2014 guidelines. J Extracell Vesicles. 2018;7(1):1535750.

Why do the authors keep writing about exosomes instead extracellular vesicles?

We thank the reviewer for the informative comment. We have updated the terminology used in our manuscript in accordance with MISEV2018. All mentions of “exosomes” have been changed to “extracellular vesicles.”

4.The figure 4B seems very satured. Could the authors show a new one?

We agree with the reviewer that the Western blots in Figure 4B for FGF2 and FGF23 are oversaturated. We have updated the figure with less saturated versions of the blot.

5.Page 10, line 370. The authors write morality. Is it correct? I think that the authors wanted to write mortality.

We thank the reviewer for catching this error. We did indeed mean mortality, which has now been updated in the text.

Round 2

Reviewer 2 Report

the authors have addressed my concerns, no more questions.